Inflammation-based prognostic scores predict the prognosis of locally advanced cervical esophageal squamous cell carcinoma patients receiving curative concurrent chemoradiotherapy: a propensity score-matched analysis

Wu Chia-Che 1
Li Shau-Hsuan 1
Lu Hung-I 2
Lo Chien-Ming 2
Wang Yu-Ming 3
Chou Shang-Yu 3
Chen Yen-Hao alex2999@cgmh.org.tw alex8701125@gmail.com 1 4 5
1 Department of Hematology-Oncology, Kaohsiung Chang Gung Memorial Hospital and Chang Gung University College of Medicine , Kaohsiung , Taiwan
2 Department of Thoracic & Cardiovascular Surgery, Kaohsiung Chang Gung Memorial Hospital and Chang Gung University College of Medicine , Kaohsiung , Taiwan
3 Department of Radiation Oncology, Kaohsiung Chang Gung Memorial Hospital and Chang Gung University College of Medicine , Kaohsiung , Taiwan
4 Graduate Institute of Clinical Medical Sciences, College of Medicine, Chang Gung University , Taoyuan , Taiwan
5 School of Medicine, Chung Shan Medical University , Taichung , Taiwan
Küçük Can
Electronic publication date: 2018 Sep 19
Publication date: 2018
Volume: 6
Electronic Location ID: e5655
Received 2018 Jul 9; Accepted 2018 Aug 28
Copyright: ©2018 Wu et al.
Copyright year: 2018
Copyright holder: Wu et al.
License: This is an open access article distributed under the terms of the Creative Commons Attribution License, which permits unrestricted use, distribution, reproduction and adaptation in any medium and for any purpose provided that it is properly attributed. For attribution, the original author(s), title, publication source (PeerJ) and either DOI or URL of the article must be cited.
License URL: https://creativecommons.org/licenses/by/4.0/

Keywords: Esophageal cancer, Concurrent chemoradiotherapy, Inflammation, Squamous cell carcinoma, Propensity score matching

Funding: National Science Council, Taiwan MOST 105-2314-B-182A-029 MOST 106-2314-B-182A-159-MY3 MOST 106-2320-B-182A-015 MOST 107-2314-B-182A-156-MY3 Chang Gung Memorial Hospital CMRPG8G0201 CMRPG8G0891 CMRPG8E153 This work was supported by grants from the National Science Council, Taiwan (MOST 105-2314-B-182A-029, MOST 106-2314-B-182A-159-MY3, MOST 106-2320-B-182A-015 and MOST 107-2314-B-182A-156-MY3), and Chang Gung Memorial Hospital (CMRPG8G0201, CMRPG8G0891, and CMRPG8E1533). There was no additional external funding received for this study. The funders had no role in study design, data collection and analysis, decision to publish, or preparation of the manuscript.

==============================
Introduction

The present study investigated the crucial role of inflammation-based prognostic scores in locally advanced cervical esophageal squamous cell carcinoma (ESCC) patients who underwent curative concurrent chemoradiotherapy (CCRT).

Methods

There were 411 ESCC patients enrolled, including 63 cervical ESCC patients. Using the propensity score matching method, 63 thoracic ESCC patients were matched to the 63 cervical ESCC patients. The inflammation-based prognostic scores included the neutrophil lymphocyte ratio (NLR), platelet lymphocyte ratio (PLR), albumin level, c-reactive protein (CRP) level, modified Glasgow prognostic score (mGPS), and CRP/albumin ratio. The chi-square test and Kaplan–Meier method were used for categorical variable data and overall survival, respectively. A Cox regression model was performed for univariate and multivariable analyses.

Results

With respect to cervical ESCC, NLR ≥ 2.5 (P = 0.019), PLR ≥ 103 (P = 0.013), CRP value >10 mg/l (P = 0.040), mGPS ≥ 1 (P = 0.040), and CRP/albumin ratio ≥ 9.5 (P = 0.033) were significant predictors of worse overall survival (OS) in the univariate analysis. In a multivariable analysis, PLR ≥ 103 (P = 0.010, HR: 2.66, 95% CI [1.27–5.58]) and mGPS ≥ 1 (P = 0.030, HR: 2.03, 95% CI [1.07–3.86]) were the independent prognostic parameters of worse OS. The prognostic value of these biomarkers in the matched thoracic ESCC patients was similar and compatible with the results in the cervical ESCC group in the univariate and multivariable analyses.

Conclusions

Our study suggests that these inflammation-based prognostic scores are helpful in clinical practice, and PLR and mGPS may predict the prognosis for locally advanced cervical ESCC patients who receive curative CCRT.

Introduction

Esophageal cancer is one of the most fatal human malignancies worldwide. In Taiwan, esophageal squamous cell carcinoma (ESCC) is the major pathologic type of esophageal cancer, accounting for more than 90% of all cases, and is the ninth leading cause of cancer-related deaths (National Department of Health, Republic of China, 2015). The cervical esophagus is a small portion of the esophagus with a length of 5 cm, and cervical ESCC accounts for only a small portion, specifically, less than 10%, of all esophageal cancer cases (Yin et al., 1983). In the past, the standard treatment for cervical ESCC was radical surgery, radiotherapy, or a combination of both. However, the surgery usually consisted of laryngoesophagectomy and reconstruction with gastric transposition or colon graft. Moreover, even with such surgery, the 5-year survival rate was only 12% to 27% and the post-operative mortality rate was high at 6% to 20% with significant morbidities (Grass et al., 2015). Recently, however, several studies have shown that concurrent chemoradiotherapy (CCRT) improves survival rates for ESCC and head/neck cancer patients; therefore, some physicians preferred definitive CCRT rather than surgical resection for cervical ESCC patients in clinical practice, especially for locally advanced status (Cooper et al., 1999; Pignon et al., 2009).

Growing evidences have revealed that inflammation plays an important role in tumor cell proliferation, migration, invasion, and metastasis, as well as disease progression (Balkwill & Mantovani, 2001; Mantovani et al., 2008). A series of inflammatory biomarkers, such as the neutrophil lymphocyte ratio (NLR), platelet lymphocyte ratio (PLR), albumin level, c-reactive protein (CRP) level, modified Glasgow prognostic score (mGPS), and CRP/albumin ratio, have been identified to predict clinical outcomes in several cancer types, including esophageal cancer (Feng, Huang & Chen, 2014; Lindenmann et al., 2014; Pinato et al., 2014; Stotz et al., 2013; Templeton et al., 2014; Yodying et al., 2016).

With respect to esophageal cancer, these biomarkers were reported to be associated with tumor progression and prognosis in esophageal cancer patients who receive different therapeutic modalities, including surgical resection, neoadjuvant chemoradiotherapy followed by esophagectomy, esophagectomy followed by adjuvant chemoradiotherapy, and definitive chemoradiotherapy (Dutta et al., 2011; Feng, Huang & Liu, 2013; Miyata et al., 2011; Sharaiha et al., 2011; Yoo et al., 2014). However, to the best of our knowledge, these biomarkers have not been evaluated in cervical ESCC patients who receive curative CCRT.

In the present study, the locally advanced cervical ESCC patients who received curative CCRT in our hospital were retrospectively reviewed, and the aim of the study was to determine the clinical impact of inflammation-based prognostic scores in locally advanced cervical ESCC patients who have undergone curative CCRT.

Material and Methods

Patient population

Study approval was obtained from the Chang Gung Medical Foundation Institutional Review Board (201800845B0), and written informed consent from the patients or their families was not judged necessary for this kind of retrospective study. We retrospectively reviewed ESCC patients with available medical records who underwent treatment between January 2005 and December 2015 at Kaohsiung Chang Gung Memorial Hospital. The eligibility criteria were as follows: (1) squamous cell carcinoma in histology; (2) locally advanced status, stage III, without distant metastasis or neck/celiac lymph node metastasis; (3) complete CCRT with curative intent; (4) survive more than 3 months after completing CCRT; (5) no history of second primary malignancy, such as head and neck cancers; (6) no form of any acute or chronic infection/inflammatory disease; and (7) Eastern Cooperative Oncology Group performance status 0–1. Ultimately, there were 411 ESCC patients who met the criteria for further analysis, including 63 patients who had tumors located in the cervical esophagus and 348 patients with thoracic esophageal tumors.

In order to prevent selection bias for better comparison, the propensity score matching method was used among the 348 thoracic ESCC patients. First, we used binary logistic regression to calculate a propensity score, with covariates including age, gender, tumor T status, tumor N status, tumor stage, and tumor grade being entered into the propensity model. Subsequently, a one-to-one match with the closest matching scores between the 63 cervical thoracic patients and 63 thoracic ESCC patients was determined. The algorithm used is shown in Fig. 1.

Figure 1 Algorithm for identifying locally advanced cervical and thoracic esophageal squamous cell carcinoma (ESCC) patients.

Definition of inflammatory biomarkers and clinical tumor stage

In our study, chest computed tomography (CT), endoscopic ultrasonography (EUS), and positron emission tomography (PET) scans were performed for each patient, and the clinical tumor stage was determined according to the 7th American Joint Committee on Cancer (AJCC) staging system (Edge et al., 2010). The definition of cervical esophageal cancer is that the tumor lies in the neck and is bordered superiorly by the hypopharynx and inferiorly by the thoracic inlet (sternal notch), approximately 15–20 cm from the incisors (Edge et al., 2010).

Blood samples were obtained before treatment to measure the biomarkers of interest, which included the white blood cell count, platelet count, neutrophil count, lymphocyte count, albumin level, and CRP level. The NLR was calculated by dividing the neutrophil count by the lymphocyte count, and the PLR was defined as the platelet count divided by the lymphocyte count; the cut-off values for the NLR and PLR were 2.5 and 103, respectively (Dutta et al., 2011; Xie et al., 2016). The cut-off levels for CRP and albumin in this study were 10 mg/l and 3.5 g/dl, respectively, with these levels being based on those used in previous studies (Forrest et al., 2003; McMillan, 2008; McMillan, 2013; McMillan et al., 2001). The mGPS was calculated using the CRP and albumin values and the scoring system was as follows: (1) patients with a normal CRP value (≤10 mg/l) were allocated a score of 0, regardless of the albumin level; (2) patients with a CRP level >10 mg/l combined with an albumin level ≥3.5 g/dl were allocated a score of 1; (3) patients with a CRP >10 mg/l and an albumin <3.5 g/dl were allocated a score of 2 (McMillan, 2008). The optimal cut-off level for the CRP/albumin ratio was defined as 9.5 in the subsequent analysis (Wei et al., 2015). All the indicators involved in the calculation of the inflammation-based prognostic scores were examined before the patients underwent chemotherapy and radiotherapy.

CCRT setting

Curative intent radiotherapy was planned for each patient, and the details are described below. First, a customized thermoplastic immobilization device was designed. Subsequently, CT-simulation for image acquisition was performed. Inverse plan intensity-modulated radiotherapy (IMRT) was then used to deliver 6- or 10-MV photons to cover the treatment field, including the neck and mediastinum. The gross target volume (GTV) was defined as the gross tumor and gross lymph nodes (LNs) according to chest CT scan and/or PET-CT images. The clinical target volume (CTV) comprehensively covered the esophagus, the mediastinal LNs, and the supraclavicular LNs. The planning target volume (PTV) was defined as the CTV expanded by 0.5–1.0 cm margins in all directions. The total doses prescribed to the PTV were 66–70 Gy in 33–35 daily fractions for cervical ESCC and 50–50.4 Gy in 25–28 daily fractions for thoracic ESCC, followed by a boost dose to the gross neck LNs for an additional 10–16 Gy in 5–8 daily fractions.

Chemotherapy was performed concurrently with radiotherapy and consisted of cisplatin (75 mg/m2; 4-hour infusion) on day 1 and 5-fluorouracil (1,000 mg/m2; continuous infusion) on days 1–4 every 4 weeks. For patients with creatinine clearance <60 mL/min, carboplatin was used instead of cisplatin (Chen et al., 2017a; Chen et al., 2017b; Chen et al., 2018).

Statistical analysis

Comparisons between the groups were performed using the chi-square test for categorical variable data. A Cox regression model was used for univariate and multivariable analyses, and the hazard ratio (HR) and 95 confidence interval (CI) were computed with the Cox proportional hazards model. Overall survival (OS) was calculated from the date of diagnosis of the esophageal cancer to the date of death or last contact. The Kaplan–Meier method was used to estimate OS, and the log rank test was performed to evaluate the differences between the groups for univariate analysis. The statistical analyses were performed with the SPSS 19 software package (IBM, Armonk, NY, USA). All of the tests were two-sided tests, and P < 0.05 was considered statistically significant.

Results

Patient characteristics

There were 411 locally advanced inoperable ESCC patients who completed CCRT with a curative intent at Kaohsiung Chang Gung Memorial Hospital who were retrospectively investigated in this study, including 63 cervical ESCC patients. All 63 cervical ESCC patients had an Eastern Cooperative Oncology Group performance status ≤1, and these patients consisted of 61 male patients and two female patients with a median age of 58 years (range, 37–80 years). The tumor T status was revealed to be T2 in 1 patient (1.6%), T3 in 13 patients (20.6%), and T4 in 49 patients (77.8%). Meanwhile, five patients (7.9%) were diagnosed as having N0 status, 24 patients (38.1%) were diagnosed as having as having N1 status, 23 patients (36.5%) were diagnosed as having as having N2 status, and 11 patients (17.5%) were diagnosed as having as having N3 status. In terms of tumor stage, seven patients (11.1%) had stage IIIA, five patients (7.9%) had stage IIIB, and 51 patients (81.0%) had stage IIIC. Among the 63 patients, 14 patients (22.3%) were classified as grade 1, 36 patients (57.1%) as grade 2, and 13 patients (20.6%) as grade 3.

Using the propensity score matching method, 63 matched patients of the 348 thoracic ESCC patients were identified to compare to the 63 cervical ESCC patients. Parameters between these two groups were all matched without statistical difference, including tumor age, gender, T status, N status, tumor stage and tumor grade. The clinicopathological characteristics of these cervical and thoracic ESCC patients are shown in Table 1.

Table 1 Clinicopathological parameters in 126 locally advanced cervical/thoracic esophageal SCC patients receiving curative CCRT.

Characteristics	aCervical esophageal SCC group (N = 63)	aThoracic esophageal SCC group (N = 63)	P value	
Age				
<60 years	37 (58.7%)	37 (58.7%)	1.0	
≥60 years	26 (41.3%)	26 (41.3%)		
Sex				
Male	61 (96.8%)	61 (96.8%)	1.0	
Female	2 (3.2%)	2 (3.2%)		
T status				
2	1 (1.6%)	1 (1.6%)	1.0	
3	13 (20.6%)	13 (20.6%)		
4	49 (77.8%)	49 (77.8%)		
N status				
0	5 (7.9%)	5 (7.9%)	1.0	
1	24 (38.1%)	24 (38.1%)		
2	23 (36.5%)	23 (36.5%)		
3	11 (17.5%)	11 (17.5%)		
Stage				
IIIA	7 (11.1%)	7 (11.1%)	1.0	
IIIB	5 (7.9%)	5 (7.9%)		
IIIC	51 (81.0%)	51 (81.0%)		
Grade				
1	14 (22.3%)	14 (22.3%)	1.0	
2	36 (57.1%)	36 (57.1%)		
3	13 (20.6%)	13 (20.6%)		
	
Notes.

SCC squamous cell carcinoma

CCRT concurrent chemoradiotherapy

a Using propensity score matching method.

* Statistically significant.

Inflammation-based prognostic scores and clinical outcomes

The inflammation-based prognostic scores used in our study included NLR, PLR, albumin value, CRP value, mGPS, and CRP/albumin ratio which were all well investigated in previous studies (Dutta et al., 2011; Feng, Huang & Chen, 2014; Forrest et al., 2003; McMillan, 2013; Miyata et al., 2011; Wei et al., 2015; Xie et al., 2016; Yodying et al., 2016). A total of 63 thoracic ESCC patients were identified to completely match the 63 cervical ESCC patients using the propensity score matching method. There were no significant differences in the baseline characteristics of these two groups, except for PLR (P = 0.047). The comparison of inflammatory biomarkers in these cervical and thoracic ESCC patients are shown in Table 2.

Table 2 Comparison of inflammation-based prognostic scores in 126 locally advanced cervical/thoracic esophageal SCC patients receiving curative CCRT.

Characteristics	aCervical esophageal SCC group (N = 63)	aThoracic esophageal SCC group (N = 63)	P value	
Neutrophil lymphocyte ratio				
<2.5	28 (44%)	19 (30%)	0.097	
≥2.5	35 (56%)	44 (70%)		
Platelet lymphocyte ratio				
<103	42 (67%)	31 (49%)	0.047*	
≥103	21 (33%)	32 (51%)		
Albumin				
<3.5	9 (14%)	12 (19%)	0.473	
≥3.5	54 (86%)	51 (81%)		
CRP				
≤ 10	28 (44%)	21 (33%)	0.201	
>10	35 (56%)	42 (67%)		
mGPS				
0	28 (44%)	21 (33%)	0.201	
1 + 2	35 (56%)	42 (67%)		
CRP/Albumin ratio				
<9.5	35 (56%)	34 (54%)	0.858	
≥9.5	28 (44%)	29 (46%)		
Notes.

SCC squamous cell carcinoma

CCRT concurrent chemoradiotherapy

CRP C-reactive protein

mGPS modified Glasgow Prognostic Score

a Using propensity score matching method.

* Statistically significant.

With respect to cervical ESCC, NLR ≥ 2.5 (P = 0.019), PLR ≥ 103 (P = 0.013), CRP value >10 (P = 0.037), mGPS ≥ 1 (P = 0.037), and CRP/albumin ratio ≥ 9.5 (P = 0.033) were significant predictors of worse OS in the univariate analysis. Patients with NLR ≥ 2.5 had worse OS compared to others with NLR <2.5 (12.0 months versus 32.6 months, P = 0.016, Fig. 2A); and worse OS (11.6 months versus 25.3 months, P = 0.010, Fig. 2B) was also found in the 21 patients with PLR ≥ 103 than the other 42 patients with PLR <103. The 35 patients with mGPS ≥ 1 were found to have worse OS in comparison with the 28 patients with mGPS of 0 (12.0 months versus 25.3 months, P = 0.037, Fig. 2C); the total of 28 patients who had CRP/albumin ratio ≥ 9.5 had worse OS compared to the other 35 patients who had CRP/albumin ratio <9.5 (12.0 months versus 25.3 months, P = 0.030, Fig. 2D). In a multivariable analysis, PLR ≥ 103 (P = 0.010, HR: 2.66, 95% CI [1.27–5.58]) and mGPS ≥ 1 (P = 0.030, HR: 2.03, 95% CI [1.07–3.86]) were the independent prognostic parameters of worse OS.

Figure 2 Comparison of overall survival curves of cervical esophageal squamous cell carcinoma patients according to different inflammation-based prognostic scores.

(A) Neutrophil lymphocyte ratio (NLR). (B) Platelet lymphocyte ratio (PLR) (C) Modified Glasgow prognostic score (mGPS). (D) CRP/albumin ratio.

With respect to thoracic ESCC, the univariate analysis showed that NLR ≥ 2.5 (P = 0.041), PLR ≥ 103 (P = 0.024), CRP value >10 (P = 0.001), mGPS ≥ 1 (P = 0.001), and CRP/albumin ratio ≥ 9.5 (P = 0.002) were still significant predictors of worse OS, similar to the results in the cervical ESCC group. Worse OS (9.0 months versus 14.4 months, P = 0.038, Fig. 3A) was also found in the 35 patients with ≥ 2.5 than the other 28 patients with NLR <2.5; and patients with PLR ≥ 103 had worse OS compared to others with PLR <103 (9.0 months versus 10.9 months, P = 0.022, Fig. 3B). The total of 35 patients who had mGPS ≥ 1 had worse OS compared to the other 28 patients with mGPS of 0 (8.9 months versus 20.0 months, P = 0.001, Fig. 3C); the 28 patients who had CRP/albumin ratio ≥ 9.5 were found to have worse OS in comparison with the 35 patients who had CRP/albumin ratio <9.5 (9.0 months versus 15.9 months, P = 0.002, Fig. 3D). Patients with NLR ≥ 2.5 (P = 0.012, HR: 2.21, 95% CI [1.19–4.12]) and mGPS ≥ 1 (P < 0.001, HR: 3.13, 95% CI [1.66–5.88]) had worse OS than others with NLR <2.5 and mGPS of 0 in the multivariable analysis. These univariate and multivariable survival analyses are shown in Table 3.

Figure 3 Comparison of overall survival curves of thoracic esophageal squamous cell carcinoma patients according to different inflammation-based prognostic scores.

(A) Neutrophil lymphocyte ratio (NLR). (B) Platelet lymphocyte ratio (PLR) (C) Modified Glasgow prognostic score (mGPS). (D) CRP/albumin ratio.

Table 3 Univariate and multivariable analysis of overall survival in 126 locally advanced cervical/thoracic esophageal SCC patients receiving curative CCRT.

Characteristics	  Univariate analysis	Multivariable analysis	
	HR (95% CI)	P value	HR (95% CI)	P value	
aCervical esophageal SCC group	
Neutrophil lymphocyte ratio ≥ 2.5	2.19 (1.14–4.20)	0.019*			
Platelet lymphocyte ratio ≥ 103	2.51 (1.22–5.17)	0.013*	2.66 (1.27–5.58)	0.010*	
Albumin ≥ 3.5	0.70 (0.31–1.61)	0.404			
CRP >10	1.95 (1.03–3.69)	0.037*			
mGPS ≥ 1	1.95 (1.03–3.69)	0.037*	2.03 (1.07–3.86)	0.030*	
CRP/Albumin ratio ≥ 9.5	1.96 (1.06–3.64)	0.033*			
aThoracic esophageal SCC group	
Neutrophil lymphocyte ratio ≥ 2.5	1.87 (1.03–3.42)	0.041*	2.21 (1.19–4.12)	0.012*	
Platelet lymphocyte ratio ≥ 103	1.89 (1.09–3.28)	0.024*			
Albumin ≥ 3.5	0.68 (0.35–1.32)	0.246			
CRP >10	2.78 (1.50–5.15)	0.001*			
mGPS ≥ 1	2.78 (1.50–5.15)	0.001*	3.13 (1.66–5.88)	<0.001*	
CRP/Albumin ratio ≥9.5	2.42 (1.38–3.24)	0.002*			
Notes.

CCRT concurrent chemoradiotherapy

HR hazard ratio

CI confidence interval

CRP C-reactive protein

mGPS modified Glasgow Prognostic Score

a Using propensity score matching method.

* Statistically significant.

Discussion

Cervical ESCC is a small population of all esophageal cancer but is often locally advanced with nearby tissue invasion at initial presentation (Yin et al., 1983). Although cervical ESCC only accounts for less than 10% of all cases, the management of this rare disease is very challenging. In the past, radical surgery with reconstruction was the gold standard of treatment, but the locally advanced status usually increased the difficulty of surgery, resulting in high mortality and morbidities (Grass et al., 2015). In recent years, growing evidences have demonstrated that CCRT improved overall survival for ESCC patients, and more and more physicians preferred definitive CCRT rather than surgical resection for cervical ESCC patients in clinical practice (Cooper et al., 1999; Pignon et al., 2009). As far as we know, there were limited studies which focused on the outcome of cervical ESCC patients, and the predictive prognostic biomarkers for this group were not available. However, accumulating evidences have revealed that inflammation-based prognostic scores, such as NLR, PLR, mGPS, etc., were associated with clinical outcome in several cancer types, including esophageal cancer (Feng, Huang & Chen, 2014; Lindenmann et al., 2014; Pinato et al., 2014; Stotz et al., 2013; Templeton et al., 2014; Yodying et al., 2016). Therefore, the current study is designed to determine the role of these inflammation-based biomarkers in cervical ESCC patients.

Systemic inflammation plays an important role in the tumorigenesis, and the mechanism is very complicated. It may cause genetic mutations and instability, suppress antitumor immunity, decrease DNA repair function, and promote the formation of microenvironments, contributing to tumor initiation. In addition, it also induces tumor cell invasion, migration, metastasis and angiogenesis, resulting in tumor progression. The inflammatory factors are mainly derived from the secretion of both host and tumor cells, and the systemic reaction to cancer cells, including some chemokines and cytokines, transcription factors, CRP, circulating immunocytes, and so on (Balkwill, 2012; Elinav et al., 2013; Hoesel & Schmid, 2013; Nimptsch et al., 2015).

In the present study, we selected six biomarkers to evaluate the prognosis in the cervical ESCC patients, including NLR, PLR, CRP, albumin, mGPS and CRP/albumin ratio. Neutrophils play an important role in the systemic inflammation, and platelets are a critical source of chemokines/cytokines, and they both promote tumor progression through many different pathways (Balkwill & Mantovani, 2001; Grivennikov, Greten & Karin, 2010). NLR and PLR were well examined to predict the prognosis and have been frequently used in clinical practice in several cancer types (Feng, Huang & Liu, 2013; Stotz et al., 2013; Templeton et al., 2014). CRP is a protein of acute phase inflammation and has been reported to be associated with prognosis in esophageal cancer patients (Nozoe, Saeki & Sugimachi, 2001). Albumin level is a good tool to evaluate the nutrition status in cancer patients, and malnutrition is strongly correlated with worse treatment response and poor prognosis (Hu et al., 2009). The scoring system of mGPS, reported by McMillan, has been reported to be related to tumor size, lymph node metastasis, degree of tumor invasion, and overall survival in ESCC patients (McMillan, 2008). The CRP/albumin ratio was initially developed to predict clinical outcome and complications in patients with severe medical illness, such as sepsis; after that, it was also indicated to predict prognosis in some cancer patients (Ranzani et al., 2013; Wei et al., 2015). These biomarkers, such as complete blood count/differential count, CRP, and albumin, are easy-to-measure, and most are usually considered as routine pre-treatment tests in clinical practice.

The goal of treatment between thoracic and cervical ESCC is a little different. For cervical ESCC patients, definitive CCRT is the more preferred therapeutic modality rather than surgical resection due to high mortality and morbidities; therefore, higher radiotherapy dose (66–70 Gy) was planned for these patients. In contrast, the radiotherapy dose for thoracic ESCC patients who received CCRT is around 50 Gy. The different radiotherapy doses may result in different treatment response of CCRT and prognosis; although there were limited evidences to discuss the issue. Therefore, the role of inflammation-based prognostic scores for cervical and thoracic ESCC may be different. In previous studies, these biomarkers were well investigated and defined as prognostic factors in thoracic ESCC patients; however, the crucial role in cervical ESCC patients was unclear (Dutta et al., 2011; Feng, Huang & Chen, 2014; McMillan, 2008; Miyata et al., 2011; Wei et al., 2015; Xie et al., 2016; Yodying et al., 2016). The current study showed that NLR, PLR, CRP, mGPS and CRP/albumin ratio were strongly correlated to poor prognosis in cervical ESCC patients. Moreover, in order to correct for bias, the propensity score matching method was used to select a control group from among the locally advanced thoracic ESCC patients who received curative CCRT according to clinical tumor parameters (TNM stage, grade, sex and age) in our hospital. The prognostic value of these biomarkers in the matched thoracic ESCC patients was similar and compatible to the results in the cervical ESCC group in the univariate and multivariable analyses.

Several studies have revealed that sex is an independent prognostic factor of overall survival in esophageal cancer (Chen et al., 2013; Cheng et al., 2018; Micheli et al., 2009). In Taiwan, the male/female incidence ratio of esophageal cancer was 16.2, and male patients were reported to have significantly worse prognosis compared to female patients (Cheng et al., 2018). In the current study, there were only two female patients and the male/female ratio was 30.5; therefore, the effect of sex in the survival analysis may be minimal.

There are some limitations in our current study. First, the study only enrolled a small sample size, and all patients were treated at a single institution. Second, the study is a retrospective review, such that there may be selection bias. However, to be best of our knowledge, the current study is the first study to investigate the crucial role of inflammation-based prognostic scores in cervical ESCC patients. Furthermore, it comprises the largest series thus far of locally advanced cervical ESCC patients who underwent curative CCRT, and may be helpful to clarify the situation of this rare ESCC group.

Conclusions

Our study suggests that the inflammation-based prognostic scores are helpful in clinical practice, and PLR and mGPS may predict the prognosis for locally advanced cervical ESCC patients who receive curative CCRT.

Supplemental Information

Data S1 Raw data

Click here for additional data file.

We appreciated the Biostatistics Center, Kaohsiung Chang Gung Memorial Hospital for statistics work.

Additional Information and Declarations

Competing Interests

Author Contributions

Human Ethics

Data Availability

The authors declare there are no competing interests.

Chia-Che Wu and Shau-Hsuan Li performed the experiments, approved the final draft.

Hung-I Lu and Chien-Ming Lo analyzed the data, approved the final draft.

Yu-Ming Wang contributed reagents/materials/analysis tools, approved the final draft.

Shang-Yu Chou prepared figures and/or tables, approved the final draft.

Yen-Hao Chen conceived and designed the experiments, authored or reviewed drafts of the paper, approved the final draft.

The following information was supplied relating to ethical approvals (i.e., approving body and any reference numbers):

Study approval was obtained from the Chang Gung Medical Foundation Institutional Review Board (201800845B0).

The following information was supplied regarding data availability:

The raw data are provided in Data S1.

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
