# Peer review of "Inflammation-based prognostic scores predict the prognosis of locally advanced cervical esophageal squamous cell carcinoma patients receiving curative concurrent chemoradiotherapy: a propensity score-matched analysis"

_PeerJ, doi:10.7717/peerj.5655_

## Round 0.1 · original submission · Major Revisions

· Academic Editor

Major Revisions

Dear Dr. Chen,

Your manuscript needs to have a major revision.

Please point-by-point address each reviewer's questions.

In addition, the following issues need to be addressed:

1) Please delete the blank areas between lines 55-63, 98-110, 183-187, 242-247, 316-322, 326-339, 349-356, 370-373.

2) In line 70, the period punctuation mark should be placed after "(Yin et al. 1983)" . This should be corrected throughout the manuscript whenever a reference was cited.

3) Page 33, the empty page starting with line 485, should be deleted.


Best Regards,

Reviewer 1 ·

Basic reporting

In this study, the authors aim to determine the clinical impact of inflammation-based prognostic scores in locally advanced cervical esophageal squamous cell carcinoma (ESCC) patients who have undergone curative CCRT. Patient data were collected retrospectively in the Kaohsiung Chang Gung Memorial Hospital from January 2005 and December 2015. This study may be interesting for oncologists who are mainly interested in ESCC. The ethical approval and consent to participate are available in the manuscript. However, the paper needs additional works, and some minor changes are recommended before considering the paper for publication.

Experimental design

I also attached the Turnitin report for plagiarism check. The similarity index at the 33rd page of the Turnitin Originality Report was 40%, and this value exceeded acceptable level range between 15%-20%. Authors must be careful about citing papers appropriately in their study.

Validity of the findings

My concerns are as follows:

In abstract,

• In Line 38-42: Please give details for statistical methods in methods section of the abstract.
• In Line 43: Please correct “(N=0.013)” as “(P=0.013)”.

In introduction,

• In Line 67: I think the reference for this sentence is missing.

In material and methods,
 
• In Line 135 and 137: The reference of (Edge S) should be corrected as (Edge S et al. 2010).

In results,

• In Line 217: Please correct “(N=0.013)” as “(P=0.013)”.
• In Line 216: Table 2 should be refereed in line 216.

In references,

• In Line 375: This reference was not cited in the manuscript.

In Table 2 and 3,

• Please write the phrase “*Statistically significant” in the footnotes of the tables (same as Table 1).
• To be consistent, please use 2 or 3 digit for decimals of the p-values (P value 0.047 or 0.04).

In Figure 2,
• To be consistent, please use 2 or 3 digit for decimals of the p-values (P value 0.010 or 0.01).

Annotated reviews are not available for download in order to protect the identity of reviewers who chose to remain anonymous.

Reviewer 2 ·

Basic reporting

Minor language editing is required in the Dsicussion and Conclusions sections.

Experimental design

No Comment

Validity of the findings

No comment

Additional comments

The study population was overwhelmingly male. Was this particularly chosen in order to match the cervical and thoracic ESCC patients, or is it because more men are affected by ESCC than female? A couple of sentences in the Discussion explaining whether there is a bias in the data due to an overwhelming male study cohort would be beneficial.

·

Basic reporting

English editing should be done again, because grammatical errors and typos are seen.

Experimental design

Fair

Validity of the findings

Fair

Additional comments

This manuscript by Chia-Che Wu et Al. is a retrospective study to identify the prognostic significance of inflammatory biomarkers on patients with cervical esophageal squamous cell carcinoma, a relatively rare disease. The result shows many easily assessable biomarkers which relate to worse survival of these patients. There are points in which this study can be adjusted to be more beneficial.

1. The reference pattern is not standard. Please follow the journal instructions.

2. Your introduction, in line 76-77, is not quite accurate, when standard protocol to date stated clearly on which patients with cervical esophageal squamous cell carcinoma are indicated for surgery.

3. In line 135-137, 205, 289, The sentences are incorrect.

4. The eligibility criteria may need to be clarified when first criteria is the patients included are stage III. Firstly, according to 8th AJCC Cancer Staging Manual, T4 tumors are classified into stage IV. According to your data, the majority of patients included had T4 tumor stage.
Secondly, squamous cell type should be included in the criteria.

5. Other parameters that concern survival should be shown, such as ECOG status and underlying diseases, since they are confounders to the study outcome.

6. These biomarkers are easy-to-measure and can be very helpful if the comparative group are different treatment modalities.

7. The cut-off values of each biomarkers should be calculated because the cut-off values depend on specific populations.

8. The conclusion should be based on results of the analysis. By concluding all biomarkers as a predictor for cervical esophageal squamous cell carcinoma is misleading.

---

## Round 0.2 · accepted · Accept

· Academic Editor

Accept

Dear Dr. Chen,

I am glad to inform you that your manuscript is acceptable for publication after you address all the issues indicated in the attached annotated manuscript. These edits are minor and can be addressed while in Production.

Best Regards,

Reviewer 1 ·

Basic reporting

no comment

Experimental design

no comment

Validity of the findings

no comment

Additional comments

From the changes made in the revised manuscript and responses provided, this manuscript may be accepted.

Reviewer 2 ·

Basic reporting

No further comments/suggestions.

Experimental design

No further comments/suggestions.

Validity of the findings

No further comments/suggestions.

Additional comments

All concerns raised during the evaluation process have been addressed.

·

Basic reporting

Good to be enough

Experimental design

Good to be enough

Validity of the findings

Good to be enough

Additional comments

Authors approximately revised the manuscript and now, this manuscript has a value for publication.